# A neurally plausible model
# for online recognition and postdiction

**Li Kevin Wenliang**      **Maneesh Sahani**
Gatsby Computational Neuroscience Unit
University College London
London, W1T 4JG
{kevinli,maneesh}@gatsby.ucl.ac.uk

## Abstract

Humans and other animals are frequently near-optimal in their ability to integrate noisy and ambiguous sensory data to form robust percepts, which are informed both by sensory evidence and by prior experience about the causal structure of the environment. It is hypothesized that the brain establishes these structures using an internal model of how the observed patterns can be generated from relevant but unobserved causes. In dynamic environments, such integration often takes the form of *postdiction*, wherein later sensory evidence affects inferences about earlier percepts. As the brain must operate in current time, without the luxury of acausal propagation of information, how does such postdictive inference come about? Here, we propose a general framework for neural probabilistic inference in dynamic models based on the distributed distributional code (DDC) representation of uncertainty, naturally extending the underlying encoding to incorporate implicit probabilistic beliefs about both present and past. We show that, as in other uses of the DDC, an inferential model can be learned efficiently using samples from an internal model of the world. Applied to stimuli used in the context of psychophysics experiments, the framework provides an online and plausible mechanism for inference, including postdictive effects.

## 1   Introduction

The brain must process a constant stream of noisy and ambiguous sensory signals from the environment, making accurate and robust real-time perceptual inferences crucial for survival. Despite the difficult and some times ill-posed nature of the problem, many behavioral experiments suggest that humans and other animals achieve nearly Bayes-optimal performance across a range of contexts involving noise and uncertainty: e.g., when combining noisy signals across sensory modalities [1, 14, 34], making sensory decisions with consequences of unequal value [48], or inferring causal structure in the sensory environment [23].

Real-time perception in dynamical environments, referred to as filtering, is even more challenging. Beliefs about dynamical quantities must be continuously and rapidly updated on the basis of new sensory input, and very often informative sensory inputs will arrive after the time of the relevant state. Thus, perception in dynamical environments requires a combination of prediction—to ensure actions are not delayed relative to the external world—and *postdiction*—to ensure that perceptual beliefs about the past are correctly updated by subsequent sensory evidence [6, 12, 17, 20, 32, 41].

Behavioral [3, 5, 24, 31, 50] and physiological [8, 9, 15] findings suggest that the brain acquires an internal model of how relevant states of the world evolve in time, and how they give rise to the stream of sensory evidence. Recognition is then formally a process of statistical inference to form perceptual beliefs about the trajectory of latent causes given observations in time. While this type of

statistical computation over probability distributions is well understood mathematically and accounts for nearly optimal perception in experiments, it remains largely unknown how the brain carries out these computations in non-trivial but biologically relevant situations. Three key questions need to be answered: How does the brain represent probabilistic beliefs about dynamical variables? How does the representation facilitate computations such as filtering and postdiction? And how does the brain learn to perform these computations?

In this work, we introduce a neurally plausible online recognition scheme that addresses these three questions. We first review the distributed distributional code (DDC) [40, 45]: a hypothesized representation of uncertainty in the brain, which has been shown to facilitate efficient and accurate computation of probabilistic beliefs over latent causes in internal models without temporal structure. Our main contribution is to show how to extend the DDC representation, along with the associated mechanisms for computation and learning, to achieve online inference within a dynamical state model. In the proposed approach, each new observation is used to update beliefs about the latent state both at the present time *and* in the recent history—thus implementing a form of *online* postdiction.

This form of recognition accounts for perceptual illusions across different modalities [41]. We demonstrate in experiments that the proposed scheme reproduces known perceptual phenomena, including the auditory continuity illusion [6, 30], and positional smoothing associated with the flash-lag effect in vision [28, 32]. We also evaluate its performance at tracking the hidden state of a nonlinear dynamical system when receiving noisy and occluded observations.

## 2 Background: neural inference in static environments

Building on previous work [19, 40, 52], Vértes and Sahani [45] introduced the DDC Helmholtz Machine for inference in hierarchical probabilistic generative models, providing a potential substrate for feedforward recognition in static environments with noiseless rate neurons. We review this approach here. See Appendix E for discussion and experiments on the robustness of DDC-based inference in the presence of neuronal noise.

### 2.1 The distributed distributional code for uncertainty

The DDC representation of the probability distribution $q(z)$ of a random variable $Z$ is given by a population of $K_\gamma$ neurons whose firing rates $r_Z$ are equal to the expected values of their "encoding" (or tuning) functions $\{\gamma_k(z)\}_{k=1}^{K_\gamma}$ under $q(z)$:

$$r_{Z,k} := \mathbb{E}_q[\gamma_k(Z)], k \in \{1, 2, ..., K_\gamma\}. \tag{1}$$

As reviewed in Appendix A.2, if $q(z)$ belongs to a minimal exponential family ($Z$ discrete or continuous) with sufficient statistics $\gamma(z)$, then the DDC $r_Z$ is the mean parameter that uniquely specifies a distribution within the family. With a rich set of $\gamma(z)$, $q(z)$ can describe a large variety of distributions, and $r_Z$ is then a very flexible representation of uncertainty.

Many computations that depend on encoded uncertainty, in fact, require the evaluation of expected values. The DDC $r_Z$ can be used to approximate expectations with respect to $Z$ by projecting a target function into the span of the encoding functions $\gamma(z)$ and exploiting the linearity of expectations [44, 45, 47]. That is, for a target function $l(z)$:

$$l(z) \approx \sum_{k=1}^{K_\gamma} \alpha_k \gamma_k(z) = \boldsymbol{\alpha} \cdot \boldsymbol{\gamma}(z) \quad \Rightarrow \quad \mathbb{E}_q[l(z)] \approx \sum_k \alpha_k r_{Z,k} = \boldsymbol{\alpha} \cdot \boldsymbol{r_Z}, \tag{2}$$

The coefficients $\boldsymbol{\alpha}$ can be learned by fitting the left-hand equation in (2) at a set of points $\{z^{(s)}\}$. This set need not follow any particular distribution, but should "cover" the region where $q(z)l(z)$ has significant mass.

### 2.2 Amortised inference with the DDC

Let the internal generative model of a static environment be given by the distribution $p(z, x) = p(z)p(x|z)$, where $z$ is latent and $x$ is observed. Inference or recognition with a DDC involves finding the expectations that correspond to the posterior distribution $p(z|x)$ for a given $x$.

$$r^*_{Z|x} := \mathbb{E}_{p(z|x)}[\gamma(z)]. \tag{3}$$

This is a deterministic quantity given $\boldsymbol{x}$. Similar to other amortized inference schemes such as those in the Helmholtz machine [10] and variational auto-encoder [22, 38], the posterior DDC may be approximated using a recognition model, with the key difference that here, the output of the recognition model takes the form of (the mean parameters of) a flexible exponential family distribution defined by rich sufficient statistics $\boldsymbol{\gamma}(\boldsymbol{z})$, rather than the natural parameters or moments of a simple parametric distribution, such as a Gaussian.

Let the recognition model be $\boldsymbol{h}(\boldsymbol{x})$. A natural cost function for $\boldsymbol{h}$ would be

$$\mathcal{L}(\boldsymbol{h}) := \mathbb{E}_{p(\boldsymbol{x})} \|\mathbb{E}_{p(\boldsymbol{z}|\boldsymbol{x})}[\boldsymbol{\gamma}(\boldsymbol{z})] - \boldsymbol{h}(\boldsymbol{x})\|_2^2 = \mathbb{E}_{p(\boldsymbol{x})} \|\boldsymbol{r}_{Z|\boldsymbol{x}}^* - \boldsymbol{h}(\boldsymbol{x})\|_2^2. \tag{4}$$

However, we do not have access to $\boldsymbol{r}_{Z|\boldsymbol{x}}^*$ for a generic internal model. Nonetheless, Proposition 1 in Appendix A.1 shows that minimizing the following expected mean squared error (EMSE)

$$\mathcal{L}^s(\boldsymbol{h}) := \mathbb{E}_{p(\boldsymbol{x})} \mathbb{E}_{p(\boldsymbol{z}|\boldsymbol{x})} \|\boldsymbol{\gamma}(\boldsymbol{z}) - \boldsymbol{h}(\boldsymbol{x})\|_2^2 = \mathbb{E}_{p(\boldsymbol{z},\boldsymbol{x})} \|\boldsymbol{\gamma}(\boldsymbol{z}) - \boldsymbol{h}(\boldsymbol{x})\|_2^2 \tag{5}$$

also minimizes (4), and they share the same optimal solution. Thus, we define the DDC representation of the approximate posterior by

$$\boldsymbol{r}_{Z|\boldsymbol{x}} := \boldsymbol{h}^*(\boldsymbol{x}), \quad \boldsymbol{h}^* = \arg\min \mathcal{L}^s(\boldsymbol{h}) = \arg\min \mathcal{L}(\boldsymbol{h}). \tag{6}$$

Thus, minimizing (5) provides a way to train $\boldsymbol{h}$ even though the true posterior DDCs are not available.

### 2.3 Learning to infer

Sensory neurons encode features of an observation from the world $\boldsymbol{x}^{(*)}$ by tuning functions $\boldsymbol{\sigma}(\boldsymbol{x})$. The mean firing rates $\boldsymbol{\sigma}(\boldsymbol{x}^{(*)}) = \int \delta(\boldsymbol{x} - \boldsymbol{x}^{(*)}) \boldsymbol{\sigma}(\boldsymbol{x}) d\boldsymbol{x}$ can be seen as encoding a deterministic belief by DDC with basis $\boldsymbol{\sigma}(\boldsymbol{x})$. The brain then needs to learn the mapping from $\boldsymbol{\sigma}(\boldsymbol{x}^{(*)})$ to $\boldsymbol{r}_{Z|\boldsymbol{x}^*}$. For biological plausibility, we restrict the recognition model to have the form $\boldsymbol{h}(\boldsymbol{x}) = \mathbf{W} \boldsymbol{\sigma}(\boldsymbol{x})$ where $\mathbf{W}$ is a weight matrix. The EMSE in (5) can thus be minimized using the delta rule, given samples from the internal model $p$:

$$\widehat{\mathbf{W}} \leftarrow \epsilon \left[ \boldsymbol{\gamma}(\boldsymbol{z}^{(s)}) - \widehat{\mathbf{W}} \boldsymbol{\sigma}(\boldsymbol{x}^{(s)}) \right] \boldsymbol{\sigma}(\boldsymbol{x}^{(s)})^{\mathsf{T}}, \quad (\boldsymbol{z}^{(s)}, \boldsymbol{x}^{(s)}) \sim p(\boldsymbol{z}, \boldsymbol{x}) \tag{7}$$

where $\epsilon$ is a learning rate.[1] The approximation error between $\boldsymbol{r}_{Z|\boldsymbol{x}}$ computed this way and the DDC of the exact posterior in (3) can be reduced by adapting the number and form of the tuning curves $\boldsymbol{\sigma}(\boldsymbol{x})$. Furthermore, as shown in Theorem 1 in Appendix A.2, minimizing (5) with $\boldsymbol{h}(\boldsymbol{x}) = \mathbf{W} \boldsymbol{\sigma}(\boldsymbol{x})$ also minimizes the expected (under $p(\boldsymbol{x})$) Kullback-Leibler (KL) divergence $\mathrm{KL}[p(\boldsymbol{z}|\boldsymbol{x}) \| q(\boldsymbol{z}|\boldsymbol{x})]$, where $q(\boldsymbol{z}|\boldsymbol{x})$ is in the exponential family with sufficient statistics $\boldsymbol{\gamma}(\boldsymbol{z})$ and mean parameters $\mathbf{W} \boldsymbol{\sigma}(\boldsymbol{x})$. The minimum of the KL divergence with respect to $\mathbf{W}$ depends on $\boldsymbol{\gamma}(\boldsymbol{z})$, and can be further lowered by using a richer set of $\boldsymbol{\gamma}(\boldsymbol{z})$.

Thus, the quality of approximation provided by the distribution implied by $\boldsymbol{r}_{Z|\boldsymbol{x}}$ to the true posterior $p(\boldsymbol{z}|\boldsymbol{x})$ depends on three factors: (i) the divergence between $p(\boldsymbol{z}|\boldsymbol{x})$ and the optimal member of the exponential family with sufficient statistic functions $\boldsymbol{\gamma}(\boldsymbol{z})$; (ii) the difference between the optimal mean parameters $\boldsymbol{r}_{Z|\boldsymbol{x}}^*$ and the value of $\mathbf{W}^* \boldsymbol{\sigma}(\boldsymbol{x})$, where $\mathbf{W}^*$ minimizes (5); and (iii) the difference between $\mathbf{W}^*$ and $\widehat{\mathbf{W}}$ estimated from a finite number of internal samples. Indeed, it is possible for generalization error in the recognition model to yield values of $\boldsymbol{r}_{Z|\boldsymbol{x}}$ that are infeasible as means of $\boldsymbol{\gamma}(\boldsymbol{z})$, although even in this case their values may be used to approximate expectations of other functions.

## 3 Online inference in dynamic environments

### 3.1 A generic internal model of the dynamic world

We now turn to a dynamic environment, the main focus of this paper. Similar to the static setting in Section 2, an internal model of the dynamic world forms the foundation for online perception and

recognition. We assume that this internal model is stationary (time-invariant), Markovian and easy to simulate or sample, and that the latent dynamics and observation emission take a generic form as

$$\boldsymbol{z}_t = \boldsymbol{f}(\boldsymbol{z}_{t\text{-}1}, \zeta_{z,t}) \tag{8a}$$

$$\boldsymbol{x}_t = \boldsymbol{g}(\boldsymbol{z}_t, \zeta_{x,t}), \tag{8b}$$

where $\boldsymbol{f}$ and $\boldsymbol{g}$ are arbitrary functions that transform the conditioning variables and noise terms $\zeta_{\cdot,t}$. The expressions (8) imply conditional distributions $p(\boldsymbol{z}_t|\boldsymbol{z}_{t\text{-}1})$ and $p(\boldsymbol{x}_t|\boldsymbol{z}_t)$, but in this form they avoid narrow parametric assumptions while retaining ease of simulation. Next, we develop online inference using DDC for the internal model described by (8), thereby extending the inference from the static hierarchical setting of [45].

## 3.2 Dynamical encoding functions

Models of neural online inference usually seek to obtain the marginal $p(\boldsymbol{z}_t|\boldsymbol{x}_{1:t})$ [11, 42] or, in addition, the pairwise joint $p(\boldsymbol{z}_{t-1}, \boldsymbol{z}_t|\boldsymbol{x}_{1:t})$ [29]. However, postdiction requires updating *all* the latent variables $\boldsymbol{z}_{1:t}$ given each new observation $\boldsymbol{x}_t$. To represent such distributions by DDC, we introduce neurons with dynamical encoding functions $\boldsymbol{\psi}_t$, a function of $\boldsymbol{z}_{1:t}$ defined by a recurrence relationship encapsulated in a function $\boldsymbol{k}$: $\boldsymbol{\psi}_t = \boldsymbol{k}(\boldsymbol{\psi}_{t-1}, \boldsymbol{z}_t)$. In particular, we choose

$$\boldsymbol{\psi}_t = \boldsymbol{k}(\boldsymbol{\psi}_{t-1}, \boldsymbol{z}_t) = \mathbf{U}\boldsymbol{\psi}_{t-1} + [\boldsymbol{\gamma}(\boldsymbol{z}_t); \mathbf{0}], \quad \|\mathbf{U}\|_2 < 1, \tag{9}$$

where $\boldsymbol{\gamma}(\boldsymbol{z}_t) \in \mathbb{R}^{K_\gamma}$ is a static feature of $\boldsymbol{z}_t$ as in (1), and $\mathbf{U}$ is a $K_\psi \times K_\psi$, $K_\psi > K_\gamma$ random projection matrix that has maximum singular value less than 1.0 to ensure stability. $\boldsymbol{\gamma}(\boldsymbol{z}_t)$ only feeds into a subset of $\boldsymbol{\psi}_t$. The set of encoding functions $\boldsymbol{\psi}_t$ is then capable of encoding a posterior distribution of the history of latent states up to time $t$ through a DDC $\boldsymbol{r}_t := \mathbb{E}_{q(\boldsymbol{z}_{1:t}|\boldsymbol{x}_{1:t})}[\boldsymbol{\psi}_t]$. If $\boldsymbol{\psi}_t$ depends only on $\boldsymbol{z}_t$ ($\mathbf{U} = \mathbf{0}$), then the corresponding DDC represents the conventional filtering distribution. With a finite population size, the dependence of $\boldsymbol{\psi}_t$ on past states decay with duration, limited to about $K_\psi/K_\gamma$ time steps for a simple delay line structure. This limit can be extended with careful choices of $\mathbf{U}$ and $\boldsymbol{\gamma}(\cdot)$ [7, 16].

## 3.3 Learning to infer in dynamical models

The goal of recognition in this framework is to compute $\boldsymbol{r}_t$ recursively in online, combining $\boldsymbol{r}_{t\text{-}1}$ and $\boldsymbol{x}_t$. Extending the ideas of amortized inference and EMSE training introduced in Section 2, we use samples from the internal model to train a recursive recognition network to compute this posterior mean. In principle the recognition function $\boldsymbol{h}_t$ should depend on time step, to minimize:

$$\mathcal{L}_t^s(\boldsymbol{h}_t; \boldsymbol{x}_{1:t\text{-}1}) = \mathbb{E}_{p(\boldsymbol{z}_{1:t}, \boldsymbol{x}_t|\boldsymbol{x}_{1:t})}\|\boldsymbol{h}_t(\boldsymbol{x}_t; \boldsymbol{x}_{1:t\text{-}1}) - \boldsymbol{\psi}_t\|_2^2. \tag{10}$$

Unlike in (5), the expectation here is taken over a distribution conditioned on the history, which may be difficult to obtain from samples. Furthermore, the optimal $\boldsymbol{h}_t^*$ depends on $\boldsymbol{x}_{1:t\text{-}1}$. Restricting $\boldsymbol{h}_t(\boldsymbol{x}_t; \boldsymbol{x}_{1:t\text{-}1}) = \mathbf{W}_t \boldsymbol{\sigma}(\boldsymbol{x}_t)$ as in Section 2.3, the optimal $\mathbf{W}_t^*$ could be computed from $\boldsymbol{r}_{t\text{-}1}$ (summarizes $\boldsymbol{x}_{1:t\text{-}1}$), albeit not straightforwardly (see Appendix B). An alternative is to explicitly parameterize the dependence of $\boldsymbol{h}_t$ on both $\boldsymbol{r}_{t\text{-}1}$ and $\boldsymbol{x}_t$, giving a time-invariant function $\boldsymbol{h}_\phi^s(\boldsymbol{r}_{t\text{-}1}, \boldsymbol{x}_t)$, and train $\phi$ using a different loss

$$\mathcal{L}_t^s(\phi) = \mathbb{E}_{q(\boldsymbol{z}_{1:t}, \boldsymbol{x}_t, \boldsymbol{x}_{1:t\text{-}1})} \left\| \boldsymbol{h}_\phi^s(\boldsymbol{r}_{t\text{-}1}, \boldsymbol{x}_t) - \boldsymbol{\psi}_t \right\|_2^2 \tag{11}$$

where $\boldsymbol{r}_{t\text{-}1}$ depends on $\boldsymbol{x}_{1:t\text{-}1}$ through recursive filtering. After training, if $\boldsymbol{h}_{\phi^*}^s(\boldsymbol{r}_{t\text{-}1}, \cdot)$ learns the exact dependence on $\boldsymbol{r}_{t\text{-}1}$ so that it is the same as $\boldsymbol{h}_t^*(\cdot)$, then the loss in (11) is the expectation of the loss in (10) over all possible observation histories. Therefore, (11) bounds the expected loss of (10) from above; minimizing (11) ensures that (10) is minimized for any given history, and the output of $\boldsymbol{h}_{\phi^*}^s(\boldsymbol{r}_{t\text{-}1}, \boldsymbol{x}_t)$ approximates the desired DDC. Whereas technically $\phi^*$ should depend on $t$, for the stationary processes we consider here the distribution of inputs $\boldsymbol{r}_t, \boldsymbol{x}$ and outputs $\phi_t$ is time-invariant as $t \to \infty$; and so $\phi^*$ is approximately time-independent for sufficiently long sequences.

We consider two biologically plausible forms of $\boldsymbol{h}_\phi^s$:

$$\text{bilinear:} \quad \boldsymbol{h}_\mathbf{W}^{bil}(\boldsymbol{r}_{t\text{-}1}, \boldsymbol{x}_t) = \mathbf{W}(\boldsymbol{r}_{t\text{-}1} \otimes \boldsymbol{\sigma}(\boldsymbol{x}_t)), \tag{12}$$

$$\text{linear:} \quad \boldsymbol{h}_\mathbf{W}^{lin}(\boldsymbol{r}_{t\text{-}1}, \boldsymbol{x}_t) = \mathbf{W}[\boldsymbol{r}_{t\text{-}1}; \boldsymbol{\sigma}(\boldsymbol{x}_t)], \tag{13}$$

---
**Algorithm 1:** Learning to infer and postdict with temporal DDC

---
**input** : internal model $\boldsymbol{f}$, $\boldsymbol{g}$ and noise source $\zeta_{(\cdot),t}$, as in (8);

         recognition model $\boldsymbol{h}_\phi^s(\boldsymbol{r}_{t\text{-}1}, \boldsymbol{x}_t)$;

         target function $l$ on which postdictive posterior expectations are to be computed, (14);

         fixed random basis $\boldsymbol{\sigma}(\cdot)$ for $\boldsymbol{x}_t$, $\boldsymbol{\gamma}(\cdot)$ for $\boldsymbol{z}_t$ and $\boldsymbol{k}(\cdot,\cdot)$, e.g. (9);

         observations from the external world $\boldsymbol{x}_t^*$ arriving at time $t$;

Initialize internal DDCs $\{\boldsymbol{r}_0^{(s)}\}_{s=1}^S$ and latent samples $\{\boldsymbol{z}_0^{(s)}\}_{s=1}^S$ from prior $p_0(\boldsymbol{z}_0)$;

Initialize $\boldsymbol{r}_0^*$ for external observations, e.g. empirical mean of $\boldsymbol{\psi}(\boldsymbol{z}_0)$;

Initialize recognition parameters $\phi$ and readout weights $\boldsymbol{\alpha}$;

Compute recurrent feature $\boldsymbol{\psi}_0^{(s)} = [\boldsymbol{\gamma}(\boldsymbol{z}_0^{(s)}); \boldsymbol{0}], \forall s \in \{1, 2, \dots, S\}$;

**while** *Online observations come in at time $t \in \{1, 2, \dots\}$* **do**

     **Updating $\phi$ and $\boldsymbol{\alpha}$**

     **for** $s \in \{1, 2, \dots, S\}$ **do**

         Simulate $\boldsymbol{z}_t^{(s)} = \boldsymbol{f}(\boldsymbol{z}_{t\text{-}1}^{(s)}, \zeta_{z,t}^{(s)})$ and $\boldsymbol{x}_t^{(s)} = \boldsymbol{g}(\boldsymbol{z}_t^{(s)}, \zeta_{x,t}^{(s)})$, (8);

         Compute $\boldsymbol{\psi}_t^{(s)} = \boldsymbol{k}(\boldsymbol{\psi}_{t\text{-}1}^{(s)}, \boldsymbol{z}_t^{(s)})$, (9); $\boldsymbol{r}_t^{(s)} = \boldsymbol{h}_\phi(\boldsymbol{r}_{t\text{-}1}^{(s)}, \boldsymbol{x}_t^{(s)})$, e.g. (12) or (13);

     **end**

     Update $\phi$ to minimize sample version of $\mathcal{L}^s$, (11):

         bilinear (12): $\Delta W_{ijk} \propto \frac{1}{S} \sum_m (r_{t,i}^{(s)} - \psi_{t,i}^{(s)}) r_{t\text{-}1,j}^{(s)} \sigma_k(\boldsymbol{x}_t^{(s)})$;

         linear (13):    $\Delta W_{ij} \propto \frac{1}{S} \sum_m (r_{t,i}^{(s)} - \psi_{t,i}^{(s)})[\boldsymbol{r}_{t\text{-}1}^{(s)}; \boldsymbol{\sigma}(\boldsymbol{x}_t^{(s)})]_j$;

     Update $\boldsymbol{\alpha}$ to better approximate $l(\boldsymbol{z}_{t\text{-}\tau:t})$ with $\boldsymbol{\psi}_t$, e.g. by delta rule;

     **Compute posterior DDC and expectation of target function**

     $\boldsymbol{r}_t^{(*)} = \boldsymbol{h}_\phi(\boldsymbol{r}_{t\text{-}1}^{(*)}, \boldsymbol{x}_t^{(*)})$;

     $\mathbb{E}_{q(\boldsymbol{z}_{t\text{-}\tau:t}|\boldsymbol{x}_{1:t})}[l(\boldsymbol{z}_{t\text{-}\tau:t})] \approx \boldsymbol{\alpha}^\mathsf{T} \boldsymbol{r}_t^{(*)}$;

**end**

**return** : $\boldsymbol{r}_t^{(*)}$ and $\mathbb{E}_{q(\boldsymbol{z}_{t\text{-}\tau:t}|\boldsymbol{x}_{1:t})}[l(\boldsymbol{z}_{t\text{-}\tau:t})]$ at time $t \in \{1, 2, \dots\}$.

---

where $\otimes$ indicates the Kronecker product. That is, $\boldsymbol{h}_\mathbf{W}^{bil}$ maps to $\boldsymbol{r}_t$ from the outer product of $\boldsymbol{r}_{t\text{-}1}$ and $\boldsymbol{\sigma}(\boldsymbol{x}_t)$, and $\boldsymbol{h}_\mathbf{W}^{lin}$ does so from the concatenation of the two (the bilinear update is discussed further in Appendix C). Both choices allow $\mathbf{W}$ to be trained by the biologically plausible delta rule, using samples $\{(\boldsymbol{r}_t^{(s)}, \boldsymbol{z}_t^{(s)}, \boldsymbol{x}_t^{(s)})\}$. The triplets can be obtained by simulating the internal model; training samples of $\boldsymbol{r}_{t\text{-}1}^{(s)}$ are bootstrapped by applying $\boldsymbol{h}_\phi$ to the simulated $\boldsymbol{x}_{1:t}^{(s)}$.

Once we infer $\boldsymbol{r}_t$, postdictive posterior expectations (with lag $\tau$) can be found in the same way as (2).

$$\mathbb{E}_{q(\boldsymbol{z}_{t\text{-}\tau}|\boldsymbol{x}_{1:t})}[l(\boldsymbol{z}_{t\text{-}\tau})] \approx \boldsymbol{\alpha} \cdot \boldsymbol{r}_t \quad \text{where} \quad \boldsymbol{\alpha} \cdot \boldsymbol{\psi}_t \approx l(\boldsymbol{z}_{t\text{-}\tau}). \tag{14}$$

This approach to online learning for inference and postdiction in the DDC framework is summarized in Algorithm 1. The complexity of learning the recognition process scales linearly with the number of internal samples from $p$ and with $K_\psi^2 K_\sigma$ for the bilinear form (12), and with $K_\psi(K_\psi + K_\sigma)$ for the linear form (13).

## 4 Experiments

We demonstrate the effectiveness of the proposed recognition method on biologically relevant simulations.[2] For each experiment, we trained the DDC filter offline until it learned the internal model, and ran inference using fixed $\phi$ and $\boldsymbol{\alpha}$. Details of the experiments are described in Appendix D. Additional results incorporating neuronal noise are shown in Appendix E.

### 4.1 Auditory continuity illusions

In the auditory continuity illusion, the percept of a complex sound may be altered by subsequent acoustic signals. Two tone pulses separated by a silent gap are perceived to be discontinuous; however, when the gap is filled by sufficiently loud wide-band noise, listeners often report an illusory

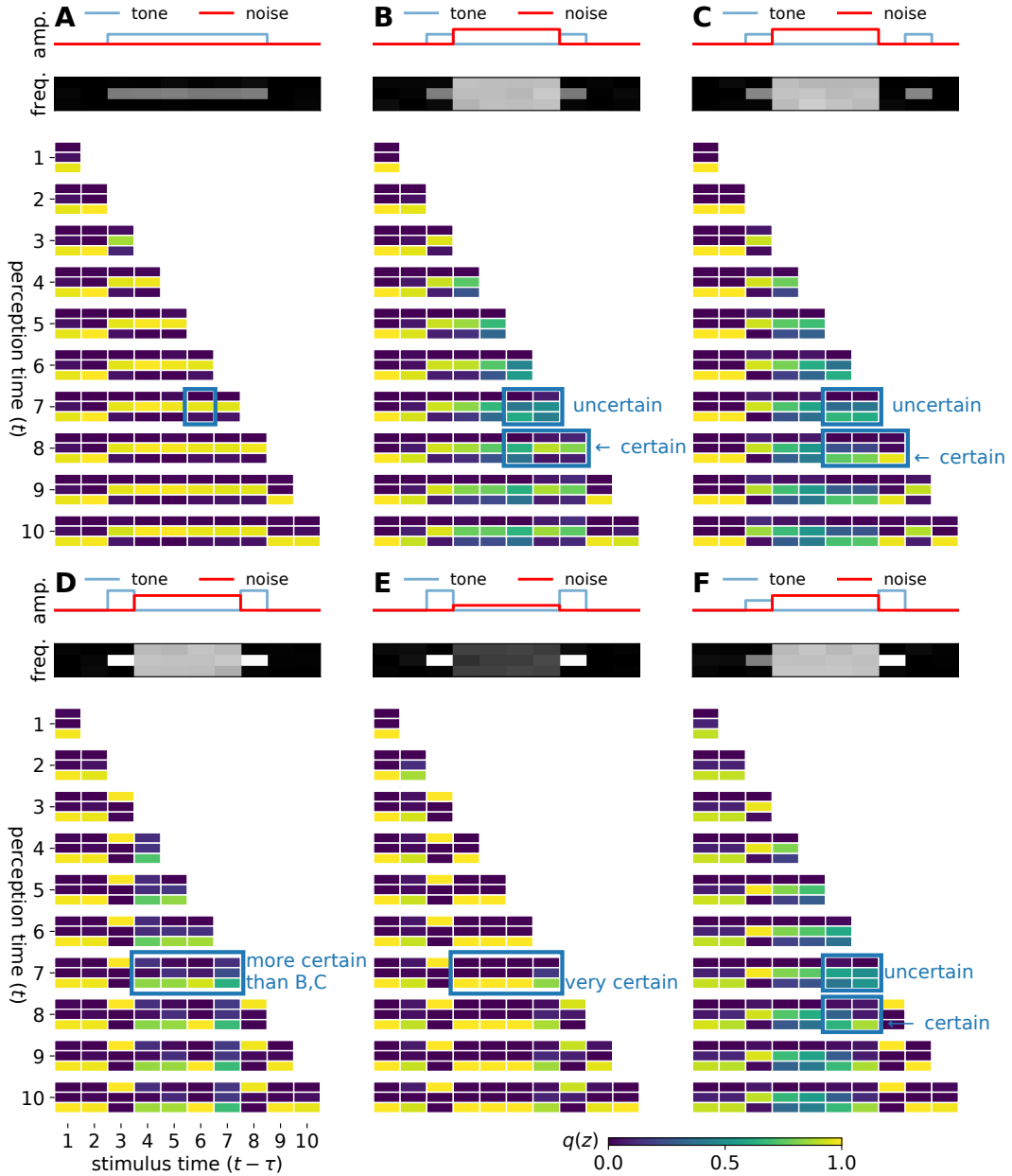

Figure 1: Modelling the auditory continuity illusion. We demonstrate postdictive DDC inference for six different acoustic stimuli (experiments A-F). In each experiment, the top panel shows the true amplitudes of the tone and noise; the middle panel shows the spectrogram observation; and the lower panel shows the real-time posterior marginal probabilities of the tone $q(\boldsymbol{z}_{t\text{-}\tau}|\boldsymbol{x}_{1:t}), \tau \in \{0,\ldots,t\text{-}1\}$ at each time $t$ and lag $\tau$. Each vertical stack of three small rectangles shows the estimated marginal probability that the tone level was zero (bottom), medium (middle) or high (top) (see scale at bottom right). Each row of stacks collects the marginal beliefs based on sensory evidence to time $t$ (left labels). The position of the stack in the row indicates the absolute time $t\text{-}\tau$ to which the belief pertains (bottom left labels). For example, the highlighted stack in A shows the marginal probability over tone level at time step 7 ($t = 7$) about the tone level at time step 6 ($t\text{-}\tau = 6$); in this example, the medium level has most of the probability as expected.

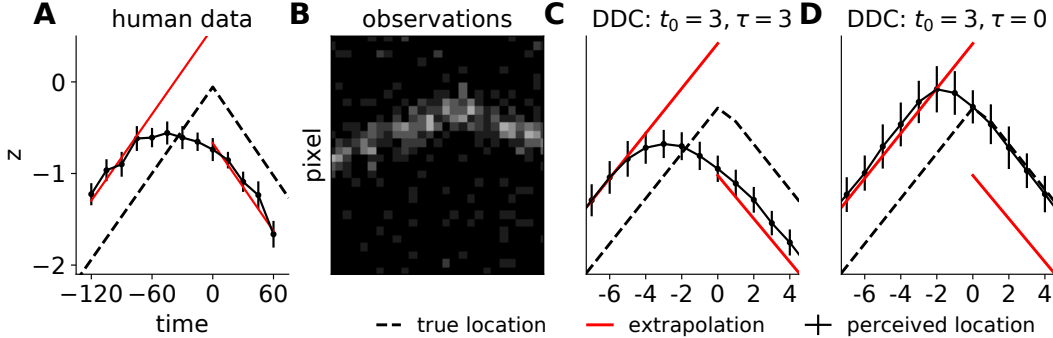

Figure 2: Modelling localization in the flash-lag effect. Black dashed line shows the true trajectory of the moving object. Red line shows the prediction of the extrapolation model. Black solid line with error bar shows the perceived trajectory reported by a human subject (mean $\pm$ 2sem) or models (mean $\pm$ std from 100 runs). A, human data from [49]. B, the observation used in our simulation. C, DDC recognition using $\tau = 3$ additional observations to postdict position at $t_0 = 3$ time steps after the time of the flash. D, DDC recognition without postdiction.

continuation of the tone through the noise. This illusion is reduced if the second tone begins after a slight delay, even though the acoustic stimulus in the two cases is identical until noise offset [6, 30].

To model the essential elements of this phenomenon, we built a simple internal model for tone and noise stimuli described in Appendix D.1, with a binary Markov chain describing the onsets and offsets of tone and wide-band noise, and noisy observations of power in three frequency bands. We ran six different experiments once the recognition model had learned to perform inference based on the internal model. Figure 1 shows the marginal posterior distributions of the perceived tone level at past times $t$-$\tau$ based on the stimulus up to time $t$, based on the DDC values $\boldsymbol{r}_t$. In Figure 1A, when a clear mid-level tone is presented, the model correctly identifies the level and duration of the tone, and retains this information following tone offset. Figure 1B and C show postdictive inference. As the noise turns on, the real-time estimate of the probability that the tone has turned off increases. However, when the noise turns off, an immediately subsequent tone restores the belief that the tone continued throughout the noise. By contrast, a gap between the noise and the second tone, increased the inferred belief that the noise had turned off to near certainty.

We tested the model on three additional sound configurations. In Figure 1D, the tone has a higher level than in Figure 1A-C. If the noise has lower spectral density than the tone, the model believes that the tone might have been interrupted, but retains some mild uncertainty. If this noise level is much lower (Figure 1E), no illusory tone is perceived. These effects of tone and noise amplitude on how likely the illusion arises are qualitatively consistent with findings in [39]. In the final experiment (Figure 1F), the model predicts that no continuity is perceived if the first tone is softer than the noise but the second tone is louder, having learned from the internal model that tone level does not, in fact, change between non-zero levels.

## 4.2 The flash-lag effect with direction reversal

In the previous experiment, the internal model correctly describes the statistics of the stimuli. It is known that a mismatch of the internal model to the real world, such as when a slowness/smooth prior meets an observation that actually moves fast [41], can induce perceptual illusions. Here, we use DDC recognition to model the flash-lag effect, although the same principle can also be used directly for the cutaneous rabbit effect in somatosensation [17].

In the flash-lag effect, a brief flash of light is generated adjacent to the current position of an object that has been moving steadily in the visual field. Subjects report the flash to appear behind the object [28, 32]. One early explanation for this finding is the extrapolation model [32]: viewers extrapolate the movement of the object and report its predicted position at the time of the flash. An alternative is the latency difference model [36] according to which the perception of a sudden flash is delayed by $t_0$ relative to the object, and so subjects report the object at time $t_0$ after the flash.

However, neither explanation can account for another related finding: if the moving object suddenly switches direction and the timing of the flash chosen at different offsets around the reversal position (still aligned with the object), the reported object locations at the time of the flashes form a smooth

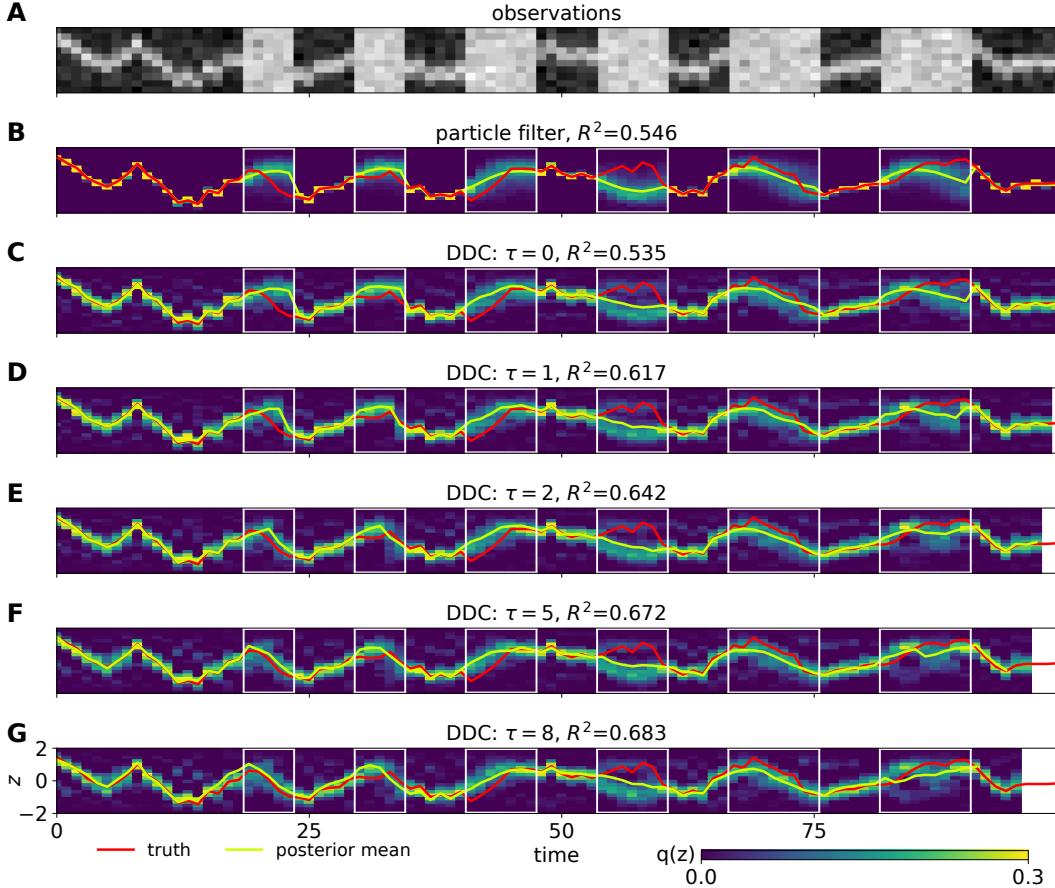

Figure 3: Tracking in a nonlinear noisy system. A, 1-D image observation through time. B, posterior mean and marginals estimated using a particle filter. C-G, posterior marginals decoded from DDC for the location at time $t$-$\tau$ perceived at time $t$.

trajectory (Figure 2A), instead of the broken line predicted by the extrapolation model, or the simple shift in time predicted by the latency difference model [49].

Rao et al. [37] suggested that the lag might arise from signal propagation delays as in the latency difference model, but the smoothing could be caused by incorporating observations during an additional processing delay. That is, after perceiving the flash at $t_0$, the brain takes time $\tau$ to estimate the object location. Importantly, subjects process more observations from the visible object trajectory in this period in order to *postdict* its position at $t_0$. The authors used Kalman smoothing in a linear Gaussian internal model favoring slow movements to reproduce the behavioral results.

Here, we apply this idea of postdiction from [37] to a more realistic internal model described in Appendix D.2. Briefly, the unobserved true object dynamics is linear Gaussian with additive Gaussian noise, and the observation emission is a 1-D image showing the position at each time step with Poisson noise (Figure 2B). After establishing a preference for slow and smooth movements, the perceived locations derived by dynamical DDC inference trace out a curve that resembles the human data, by taking into account observations after the perception of flash (Figure 2C). Without postdiction (Figure 2D), the reported location tends to overshoot, as also noted in [37].

### 4.3 Noisy and occluded tracking

When tracking a target (such as a prey) using noisy and occasionally occluded observations, it is possible to improve estimates of the trajectory followed during the occlusion by using later observations. Knowledge of the particular path followed by the target may be important for planning and control [2]. To explore the potential for dynamic DDC inference in this setting, we instantiated a system of stochastic oscillatory dynamics observed through a 1-D image with additive Gaussian

noise and occlusion (details in Appendix D.3). An example set of observations is shown in Figure 3A. We ran a simple bootstrap particle filter (PF) as a benchmark Figure 3B.

The results of DDC recognition for these observations are shown in Figure 3C-G. The marginal posterior histograms were obtained by projecting $r_t$ onto a set of bin functions using (14). (maximum entropy decoding is less smooth, see Figure 5 in Appendix D.3). We computed the $R^2$ of the prediction of true latent locations by posterior means. The purely forward ($\tau = 0$) posterior mean is comparable to that of the particle filter. As the postdictive window (and so number of future observations) $\tau$ increases, we see not only an increase in $R^2$, but also a reduction in uncertainty. In the occluded regions, the posterior mass becomes more concentrated as the number of additional observations $\tau$ increases, particularly towards the end of occlusions. In addition, bimodality is observed during some occluded intervals, reflecting the nonlinearity in the latent process.

## 5 Related work and discussion

The DDC [45] stems from earlier proposals for neural representations of uncertainty [40, 51, 52]. Notably, the DDC for a marginal distribution (1) is identical to the encoding scheme in [40], in which moments of a set of tuning functions $\gamma(z)$ encode multivariate random variables or intensity functions. The DDC may also be seen as a mean embedding within a finite-dimensional Hilbert space, approaching the full kernel mean embedding [43] as the size of the population grows. Recent developments [44, 47] focus on conditional DDCs with applications in learning hierarchical generative models, with a relationship to the conditional mean embedding [18].

The work in this paper extends the DDC framework in two ways. First, the dynamic encoding function introduced in Section 3.2 condenses information about variables at different times, and thus facilitates online postdictive inference for a generic internal model. Second, Algorithm 1 in Section 3.3 is a neurally plausible method for learning to infer. It allows a recognition model to be trained using samples and DDC messages, and could be extended to other graph structures. Although the psychophysical experiments modeled in Section 4 have been explained as smoothing on a computational level, we provides a plausible mechanism for how neural populations could implement and learn to perform this computation in an online manner.

Other schemes besides the DDC have been proposed for the neural representation of uncertainty. These include: sample-based representations [21, 25, 33]; probabilistic population codes (PPCs) [4, 27] which in their most common form have neuronal activity represent the natural parameters of an exponential family distribution [4]; linear density codes [13]; and further proposals adapted to specific inferential problems, such as filtering [11, 26]. The generative process of a realistic dynamical environment is usually nonlinear, making postdiction or even ordinary filtering challenging. If beliefs about latent states were represented by samples [25, 29], then postdiction would either depend on samples being maintained in a "buffer" to be modified by later inputs and accessed by downstream processing; would require an exponentially large number of neurons to provide samples from latent histories; or would require a complex distributed encoding of samples that might resemble the dynamic DDC we propose. Natural parameters (as in the PPC) might be associated with dynamic encoding functions as described here, but the derivation and neural implementation for the update rule would not be straightforward. In contrast, DDC (mean parameters) can be updated using simple operations as in (13) and (12). Unlike the sample-based representation hypotheses in which *posterior* samples must be drawn in real-time, sampling within the DDC learning framework is used to train the recognition model using the unconditioned joint distribution.

Although several approximate inference methods may seem plausible, learning the appropriate networks to implement them poses yet another challenge for the brain. In most of the frameworks mentioned above, special neural circuits need to be wired for specific problems. Learning to infer using DDC requires training samples from the internal model, on which the delta-rule is used to update the recognition model. This can be done off-line and does not require true posteriors as targets.

One aspect we did not address in this paper is how the brain acquires an appropriate internal model, and thus adapts to new problems. If an EM- or wake-sleep-like algorithm is used for adaptation, parameters in the internal model may be updated using the posterior representations [45] learned from the previous internal model. We expect that the postdictive (smoothed) DDC proposed here may help to fit a more accurate model to dynamical observations, as these posteriors better capture the correlations in the latent dynamics than a filtered posterior.

**Acknowledgments**

This work is supported by the Gatsby Charitable Foundation.

## Footnotes

[1] Throughout this paper we shall denote by $\boldsymbol{x}^{(*)}$ an observation from the external world, and by $\boldsymbol{x}^{(s)}$ a sample from the internal model of the world. Superscript * without parentheses indicates optimal function/parameter.

[2]Code available at `https://github.com/kevin-w-li/ddc_ssm`

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
