[Supplementary Material]

# A neurally plausible model for online recognition and postdiction:
# Supplementary material

## A  Properties of EMSE minimization

### A.1  EMSE and the expected error in estimating the DDC of exact posterior

The following proposition sets up the relationship between the EMSE in (5) and the error between $\boldsymbol{h}(\boldsymbol{x})$ and the true posterior mean (3).

**Proposition 1.** *Minimizing* (5) *minimizes* $\mathbb{E}_{p(\boldsymbol{x})}[\mathbb{E}_{p(\boldsymbol{z}|\boldsymbol{x})}[\boldsymbol{\gamma}(\boldsymbol{z})] - \boldsymbol{h}(\boldsymbol{x})\|^2]$, *the expected l-2 squared distance between the true posterior DDC and the recognition model prediction. The minimum is achieved when* $\boldsymbol{h}(\boldsymbol{x}) = \mathbb{E}_{p(\boldsymbol{z}|\boldsymbol{x})}[\boldsymbol{\gamma}(\boldsymbol{z})]$

*Proof.* Following the standard decomposition of the MSE for regression with additional expectation on $p(\boldsymbol{x})$,

$$
\begin{aligned}
(5) &= \mathbb{E}_{p(\boldsymbol{z},\boldsymbol{x})}[\|\boldsymbol{\gamma}(\boldsymbol{z}) - \boldsymbol{h}(\boldsymbol{x})\|^2] \\
&= \mathbb{E}_{p(\boldsymbol{z},\boldsymbol{x})}[\|\boldsymbol{\gamma}(\boldsymbol{z}) - \mathbb{E}_{p(\boldsymbol{z}|\boldsymbol{x})}[\boldsymbol{\gamma}(\boldsymbol{z})] + \mathbb{E}_{p(\boldsymbol{z}|\boldsymbol{x})}[\boldsymbol{\gamma}(\boldsymbol{z})] - \boldsymbol{h}(\boldsymbol{x})\|^2] \\
&= \mathbb{E}_{p(\boldsymbol{x})} \operatorname{Tr}[\mathbb{C}_{p(\boldsymbol{z}|\boldsymbol{z})}(\boldsymbol{\gamma}(\boldsymbol{z}))] + \mathbb{E}_{p(\boldsymbol{x})}[\mathbb{E}_{p(\boldsymbol{z}|\boldsymbol{x})}[\boldsymbol{\gamma}(\boldsymbol{z})] - \boldsymbol{h}(\boldsymbol{x})\|^2].
\end{aligned}
\tag{15}
$$

The cross term in the second line is zero because

$$
\mathbb{E}_{p(\boldsymbol{z},\boldsymbol{x})}[(\boldsymbol{\gamma}(\boldsymbol{z}) - \mathbb{E}_{p(\boldsymbol{z}|\boldsymbol{x})}[\boldsymbol{\gamma}(\boldsymbol{z})]) \cdot (\mathbb{E}_{p(\boldsymbol{z}|\boldsymbol{x})}[\boldsymbol{\gamma}(\boldsymbol{z})] - \boldsymbol{h}(\boldsymbol{x}))]
$$
$$
= \mathbb{E}_{p(\boldsymbol{x})}[\mathbb{E}_{p(\boldsymbol{z}|\boldsymbol{x})}[(\boldsymbol{\gamma}(\boldsymbol{z}) - \mathbb{E}_{p(\boldsymbol{z}|\boldsymbol{x})}[\boldsymbol{\gamma}(\boldsymbol{z})]] \cdot (\mathbb{E}_{p(\boldsymbol{z}|\boldsymbol{x})}[\boldsymbol{\gamma}(\boldsymbol{z})] - \boldsymbol{h}(\boldsymbol{x}))] = 0
$$

The first term is a positive constant that is independent of $\boldsymbol{h}$. The second term is minimized at 0 when $\boldsymbol{h}(\boldsymbol{x}) = \mathbb{E}_{p(\boldsymbol{z}|\boldsymbol{x})}[\boldsymbol{\gamma}(\boldsymbol{z})]$, which in turn minimizes (5). $\square$

Therefore, minimizing (5) effectively minimizes the second term of (15) which is the expected $l$-2 squared distance between the prediction $\boldsymbol{h}(\boldsymbol{x})$ and the DDC of exact posterior under $\boldsymbol{\gamma}(\boldsymbol{z})$, which depends on the flexibility of $\boldsymbol{h}$.

### A.2  MSE and the expected KL divergence

We first review a few known results for minimal exponential family from [46].

**Definition 1.** *(Minimal exponential family [46, Section 3.2]) A minimal exponential family distribution has the form*

$$
q(\boldsymbol{z}) = \exp(\boldsymbol{\theta} \cdot \boldsymbol{\gamma}(\boldsymbol{z}) - \Phi(\boldsymbol{\theta}))
\tag{16}
$$

*in which there does not exist a nonzero real vector $\boldsymbol{a}$ such that the linear combination $\boldsymbol{a} \cdot \boldsymbol{\gamma}(\boldsymbol{z})$ is equal to a constant.*

If $\boldsymbol{\gamma}$ is chosen to be a nonlinearity on random linear projections of $\boldsymbol{z}$, e.g. $\gamma_i = \tanh(\boldsymbol{v}_i \cdot \boldsymbol{z} + b)$ with elements of $\boldsymbol{v}_i$ and $b$ being draws from a random distribution, then the $\boldsymbol{\gamma}$ is linearly independent with probability one.

**Lemma 1.** *(Log normalizer derivatives [46, Proposition 3.1]) Let $\boldsymbol{r}_Z(\boldsymbol{\theta}) = \mathbb{E}[\boldsymbol{\gamma}(\boldsymbol{z})])$ be the mean parameter of a minimal exponential family distribution in* (16)*, the following holds:*

$$
\frac{\partial \Phi(\boldsymbol{\theta})}{\partial \theta_i} = r_{Z,i}(\boldsymbol{\theta}) = \mathbb{E}[\gamma_i(\boldsymbol{z})])
\tag{17}
$$

$$
\frac{\partial^2 \Phi(\boldsymbol{\theta})}{\partial \theta_i \partial \theta_j} = \frac{\partial r_{Z,i}(\boldsymbol{\theta})}{\partial \theta_j} = \mathbb{E}[\gamma_i(\boldsymbol{z})\gamma_j(\boldsymbol{z})^\intercal]) - \mathbb{E}[\gamma_i(\boldsymbol{z})]\mathbb{E}[\gamma_i(\boldsymbol{z})] =: \mathbb{C}[\boldsymbol{\gamma}(\boldsymbol{z})]_{ij}
\tag{18}
$$

Note that $\nabla_{\boldsymbol{\theta}}\Phi(\boldsymbol{\theta})$ maps from $\boldsymbol{\theta}$ to $\boldsymbol{r}$ if and only if the exponential family distribution is minimal [46, Proposition 3.2]. In addition, under the same condition, there exists a mapping $\boldsymbol{\theta}(\boldsymbol{r})$ such that $\mathbb{E}[\boldsymbol{\gamma}(\boldsymbol{z})] = \boldsymbol{r}$. Thus, the exponential family defined by the sufficient statistics $\boldsymbol{\gamma}(\boldsymbol{z})$ can be specified

by either $\boldsymbol{\theta}$ or $\boldsymbol{r}$. Importantly, $\boldsymbol{r}$ is a valid or feasible mean parameter if there exists some $q$ such that $\mathbb{E}_q[\boldsymbol{\gamma}(\boldsymbol{z})] = \boldsymbol{r}$. Thus, $\boldsymbol{\gamma}$ defines a family of distributions by the set of all feasible mean parameters.

Let an internal model take joint distribution $p(\boldsymbol{z}, \boldsymbol{x})$. Given a posterior DDC $\boldsymbol{r}(\boldsymbol{x}) = \boldsymbol{h}_\phi(\boldsymbol{x})$, let the implied (by maximum entropy) exponential family distribution be $q_\phi(\boldsymbol{z}|\boldsymbol{x}) := \exp(\boldsymbol{\theta}(\boldsymbol{r}(\boldsymbol{x})) \cdot \boldsymbol{\gamma}(\boldsymbol{z}) - \Phi(\boldsymbol{\theta}(\boldsymbol{r}(\boldsymbol{x}))))$. Let the error between the predicted and true DDC for a given $\phi$ be $\boldsymbol{e}_\phi(\boldsymbol{x}) = \boldsymbol{h}_\phi(\boldsymbol{x}) - \mathbb{E}_{p(\boldsymbol{z}|\boldsymbol{x})}[\boldsymbol{\gamma}(\boldsymbol{z})] = \boldsymbol{r}(\boldsymbol{x}) - \mathbb{E}[\boldsymbol{\gamma}(\boldsymbol{z})|\boldsymbol{x}]$.

**Theorem 1.** *Under the following assumptions:*

- $\boldsymbol{\gamma}(\boldsymbol{z})$ *forms a minimal exponential family;*

- $\boldsymbol{r}(\boldsymbol{x})$ *is a valid expectation under $q_\phi(\boldsymbol{z}|\boldsymbol{x})$ for any $\boldsymbol{x}$ ($\boldsymbol{r}(\boldsymbol{x})$ is in the set of feasible means);*

*If $\boldsymbol{e}_\phi(\boldsymbol{x}) = \boldsymbol{0}$ for some $\phi^*$ for all $\boldsymbol{x}$, then $\nabla_\phi \mathrm{KL}[p(\boldsymbol{z}|\boldsymbol{x})||q_\phi(\boldsymbol{z}|\boldsymbol{x})]\big|_{\phi^*} = \boldsymbol{0}$. Further, using $\boldsymbol{h}_\phi(\boldsymbol{x}) = \mathbf{W}\boldsymbol{\sigma}(\boldsymbol{x})$ ($\phi = \mathbf{W}$) as the recognition model, and let $\mathbf{W}^*$ be the minimizer of the EMSE problem in (5). If there exists $\epsilon_{\mathbf{W}^*} > 0$ such that $\mathbb{E}_{p(\boldsymbol{z})}[\|\boldsymbol{e}_{\mathbf{W}^*}(\boldsymbol{x})\|_2^2] \leq \epsilon_\phi^2$, and there exists an order 3 tensor $\mathbf{A}$ and $\epsilon_c > 0$ such that $\mathbb{E}_{p(\boldsymbol{x})}[\|\mathbf{e}_c(\boldsymbol{x})\|_2^2] \leq \epsilon_c^2$ where $\mathbf{e}_c(\boldsymbol{x}) = \nabla_{\mathbf{W}}\boldsymbol{\theta}(\boldsymbol{r}(\boldsymbol{x}))\big|_{\mathbf{W}^*} - \mathbf{A}\boldsymbol{\sigma}(\boldsymbol{x})$ then*

$$\left\| \mathbb{E}_{p(\boldsymbol{x})}\left[\nabla_{\mathbf{W}} \mathrm{KL}[p(\boldsymbol{z}|\boldsymbol{x})||q_{\mathbf{W}}(\boldsymbol{z}|\boldsymbol{x})]\big|_{\mathbf{W}=\mathbf{W}^*}\right]\right\|^2 \leq \epsilon_c \epsilon_\phi. \tag{19}$$

*Proof.* The proof uses the same technique to show that the Expectation Propagation algorithm with exponential family factors minimizes a similar KL. For brevity, let $\mathrm{KL}[p||q] := \mathrm{KL}[p(\boldsymbol{z}|\boldsymbol{x})||q_\phi(\boldsymbol{z}|\boldsymbol{x})]$ (note that $\mathbf{e}_c$ is a matrix)

$$\mathrm{KL}[p||q] = \int p(\boldsymbol{z}|\boldsymbol{x})\left[\log p(\boldsymbol{z}|\boldsymbol{x}) - \log q_\phi(\boldsymbol{z}|\boldsymbol{x})\right]d\boldsymbol{z}$$

$$= -\int p(\boldsymbol{z}|\boldsymbol{x})\left[\log q_\phi(\boldsymbol{z}|\boldsymbol{x})\right]d\boldsymbol{z}$$

$$\nabla_\phi \mathrm{KL}[p||q] = -\int p(\boldsymbol{z}|\boldsymbol{x})\left[\nabla_\phi\boldsymbol{\theta}(\boldsymbol{r})\boldsymbol{\gamma}(\boldsymbol{z}) - \nabla_\phi\Phi(\boldsymbol{\theta}(\boldsymbol{r}))\right]d\boldsymbol{z}$$

$$= -\int p(\boldsymbol{z}|\boldsymbol{x})\left[\nabla_\phi\boldsymbol{\theta}(\boldsymbol{r})(\boldsymbol{\gamma}(\boldsymbol{z}) - \frac{d\Phi(\boldsymbol{\theta}(\boldsymbol{r}))}{d\boldsymbol{\theta}(\boldsymbol{r})})\right]d\boldsymbol{z}$$

$$= \nabla_\phi\boldsymbol{\theta}(\boldsymbol{r})\left[\mathbb{E}[\boldsymbol{\gamma}(\boldsymbol{z})|\boldsymbol{x}] - \boldsymbol{r}\right]$$

$$= \nabla_\phi\boldsymbol{\theta}(\boldsymbol{r})\left[\boldsymbol{e}_\phi(\boldsymbol{x})\right]. \tag{20}$$

The second to last equality follows (17). Clearly, $\nabla_\phi \mathrm{KL}[p||q] = \boldsymbol{0}$ if $\boldsymbol{e}_{\mathbf{W}}(\boldsymbol{x}) = \boldsymbol{0}, \forall \boldsymbol{x}$.

Now suppose $\boldsymbol{r}(\boldsymbol{x}) = \mathbf{W}\boldsymbol{\sigma}(\boldsymbol{x})$. We decompose $\nabla_{\mathbf{W}}\boldsymbol{\theta}(\boldsymbol{r})$ as follows

$$\nabla_{\mathbf{W}}\boldsymbol{\theta}(\boldsymbol{r}(\boldsymbol{x})) = \nabla_{\mathbf{W}}\boldsymbol{\theta}(\boldsymbol{r}) - \mathbf{A}\boldsymbol{r}(\boldsymbol{x}) + \mathbf{A}\boldsymbol{r}(\boldsymbol{x}) \tag{21}$$

$$= \mathbf{e}_c(\boldsymbol{x}) + \mathbf{A}\boldsymbol{r}(\boldsymbol{x}). \tag{22}$$

Substituting in (20) and taking the expectation over $p(\boldsymbol{x})$ gives

$$\mathbb{E}[\nabla_{\mathbf{W}}\mathrm{KL}[p||q]] = \mathbb{E}\left[\mathbf{e}_c(\boldsymbol{x})\boldsymbol{e}_{\mathbf{W}}(\boldsymbol{x}) + (\mathbf{A}\boldsymbol{r}(\boldsymbol{x}))\cdot\boldsymbol{e}_{\mathbf{W}}(\boldsymbol{x})\right]$$

$$= \mathbb{E}\left[\mathbf{e}_c(\boldsymbol{x})\boldsymbol{e}_{\mathbf{W}}(\boldsymbol{x})\right] + \mathbf{A}\mathbf{W}\mathbb{E}\left[\boldsymbol{\sigma}(\boldsymbol{x})\boldsymbol{e}_{\mathbf{W}}^{\mathsf{T}}(\boldsymbol{x})\right]$$

$$\mathbb{E}[\|\nabla_{\mathbf{W}}\mathrm{KL}[p||q]\big|_{\mathbf{W}^*}\|] \overset{(1)}{\leq} \sqrt{\mathbb{E}\left[\|\mathbf{e}_c(\boldsymbol{x})\|_2^2\right]}\sqrt{\mathbb{E}\left[\|\boldsymbol{e}_{\mathbf{W}^*}(\boldsymbol{x})\|_2^2\right]} + \|\mathbf{A}\|_2^2\|\mathbf{W}^*\|_2^2\|\mathbb{E}\left[\boldsymbol{\sigma}(\boldsymbol{x})\boldsymbol{e}_{\mathbf{W}^*}^{\mathsf{T}}(\boldsymbol{x})\right]\|_2^2$$

$$\overset{(2)}{\leq} \epsilon_c \epsilon_{\mathbf{W}^*},$$

where (1) is due to the Cauchy-Schwarz inequality, and (2) is because the last term is the gradient of the EMSE w.r.t. $\mathbf{W}$, which is $\boldsymbol{0}$ when using $\mathbf{W}^*$ that solving the EMSE problem:

$$\mathbb{E}_{\boldsymbol{x}}[\boldsymbol{\sigma}(\boldsymbol{x})\mathbf{e}_{\mathbf{W}^*}^{\mathsf{T}}] = \mathbb{E}_{\boldsymbol{x}}\left[\boldsymbol{\sigma}(\boldsymbol{x})\left(\mathbb{E}_{\boldsymbol{z}|\boldsymbol{x}}\boldsymbol{\gamma}(\boldsymbol{z}) - \mathbf{W}^*\boldsymbol{\sigma}(\boldsymbol{x})\right)^{\mathsf{T}}\right] = \mathbb{E}_{\boldsymbol{z},\boldsymbol{x}}\left[\boldsymbol{\sigma}(\boldsymbol{x})\left(\boldsymbol{\gamma}(\boldsymbol{z}) - \mathbf{W}^*\boldsymbol{\sigma}(\boldsymbol{x})\right)^{\mathsf{T}}\right] = \boldsymbol{0}.$$

$\square$

The first assumption holds almost always. The second assumption is in general hard to reinforce, but after optimizing $\phi$, the DDC $\boldsymbol{r}(\boldsymbol{x})$ is likely to be inside the set of feasible means unless the true

posterior is close to a delta distribution on a single value of $\boldsymbol{z}$, in which case the true posterior mean lies close to the boundary of the feasible set, and estimation error is likely to push $\boldsymbol{r}$ out of the feasible set.

The bound in (19) suggests that whenever $\epsilon_{\mathbf{W}}$ is small, the gradient of the KL is also small. For finite independent samples from $p$, $\epsilon_{\mathbf{W}}$ shrinks at rate $1/\sqrt{n}$. The multiplier $\epsilon_c$ suggests that the gradient of KL goes to zero faster if $\boldsymbol{\sigma}(\boldsymbol{x})$ better approximates the Jacobian $\boldsymbol{\theta}(\boldsymbol{r})$ w.r.t $\boldsymbol{r}$ ($\epsilon_c$ is small). This Jacobian is, after inverting the total derivative $\frac{d\boldsymbol{r}}{d\boldsymbol{\theta}}$ and using (18), is $[\mathbb{C}_{q(\boldsymbol{x}|\boldsymbol{x})}(\boldsymbol{\gamma}(\boldsymbol{x}))]^{-1}$, which depends on the exponential family defined by $\boldsymbol{\gamma}(\boldsymbol{z})$.

Thus, Theorem 1 suggests that an ideal $\boldsymbol{\sigma}(\boldsymbol{x})$ would be rich enough to linearly approximate not just the posterior mean but also the posterior covariance of $\boldsymbol{\gamma}(\boldsymbol{z})$ for all $\boldsymbol{x}$. A simple $\boldsymbol{\gamma}(\boldsymbol{z})$ would help a given $\boldsymbol{\sigma}(\boldsymbol{x})$ satisfy these requirements, but a too simple $\boldsymbol{\gamma}(\boldsymbol{z})$ may not be rich enough to approximate a more complicated distribution, and the lowest KL could still be large even after optimizing the recognition parameters.

## B    Formal solution to the filtering loss

We show the formal solution to minimizing (10) before discussing its biological implications.

**Proposition 2.** *Given a DDC of previous belief $\boldsymbol{r}_{t\text{-}1}$, $\mathbf{W}_{\boldsymbol{r}_{t\text{-}1}}$ below is the minimizer of* (10)

$$\mathcal{L}^f(\mathbf{W}) = \mathbb{E}_{q(\boldsymbol{z}_{1:t},\boldsymbol{x}_t|\boldsymbol{x}_{1:t-1})}\left[\|\mathbf{W}\boldsymbol{\sigma}(\boldsymbol{x}_t) - \boldsymbol{\psi}(\boldsymbol{z}_{1:t})\|_2^2\right] \qquad \text{(10 revisited)}$$

$$\mathbf{W}_{\boldsymbol{r}_{t\text{-}1}} = \mathbf{C}_{\boldsymbol{Z}_{1:t},\boldsymbol{X}_t|\boldsymbol{x}_{1:t-1}}\mathbf{C}_{\boldsymbol{X}_t,\boldsymbol{X}_t|\boldsymbol{x}_{1:t-1}}^{-1} \qquad (23)$$

$$\mathbf{C}_{\boldsymbol{Z}_{1:t},\boldsymbol{X}_t|\boldsymbol{x}_{1:t-1}} = \mathbf{C}_{\boldsymbol{Z}_{1:t},\boldsymbol{X}_t|\boldsymbol{Z}_{t-1}}\boldsymbol{r}_{t\text{-}1} \quad \mathbf{C}_{\boldsymbol{X}_t,\boldsymbol{X}_t|\boldsymbol{x}_{1:t-1}} = \mathbf{C}_{\boldsymbol{X}_t,\boldsymbol{X}_t|\boldsymbol{Z}_{t-1}}\boldsymbol{r}_{t\text{-}1}$$

$$\mathbf{C}_{\boldsymbol{Z}_{1:t},\boldsymbol{X}_t|\boldsymbol{Z}_{t-1}} = \arg\min_{\mathbf{C}} \mathbb{E}_{p(\boldsymbol{z}_{t-1},\boldsymbol{z}_t,\boldsymbol{x}_t)}\|\mathbf{C}\boldsymbol{\psi}_{t\text{-}1} - \boldsymbol{\psi}(\boldsymbol{z}_{1:t})\boldsymbol{\sigma}(\boldsymbol{x}_t)^\mathsf{T}\|_2^2$$

$$\mathbf{C}_{\boldsymbol{X}_t,\boldsymbol{X}_t|\boldsymbol{Z}_{t-1}} = \arg\min_{\mathbf{C}} \mathbb{E}_{p(\boldsymbol{z}_{t-1},\boldsymbol{x}_t)}\|\mathbf{C}\boldsymbol{\psi}_{t\text{-}1} - \boldsymbol{\sigma}(\boldsymbol{x}_t)\boldsymbol{\sigma}(\boldsymbol{x}_t)^\mathsf{T}\|_2^2.$$

This is similar to the kernel Bayes rule [18]. The two minimization problems are essentially computing the readout weights used to approximate the conditional covariance matrices $\mathbf{C}$. This solution for filtering involves solving these two problems before taking an inverse of a correlation matrix. If one interprets the two tensor $\mathbf{C}$'s as weights, the matrix $\mathbf{C}$'s are readout from $\boldsymbol{r}_{t\text{-}1}$, then it is not clear how the inverse and $\mathbf{W}_{\boldsymbol{r}_{t\text{-}1}}$ could be implemented by neural mechanisms.

## C    Approximated solution for filtering

### C.1    The bilinear approximation and the tensor train decomposition

The bilinear approximation $\boldsymbol{h}_{\mathbf{W}}(\boldsymbol{r}_{t\text{-}1}, \boldsymbol{x}_t)$ (12) and the corresponding solution to minimizing the EMSE (11) w.r.t. $\mathbf{W}$ is connected to the tensor train decomposition (TT) [35]. The EMSE is

$$\mathcal{L}^{bil}(\mathbf{W}) = \mathbb{E}_{q(\boldsymbol{z}_{1:t},\boldsymbol{x}_t,\boldsymbol{x}_{1:t-1})}\left[\|\mathbf{W}\cdot(\boldsymbol{r}_{t-1}\otimes\boldsymbol{x}_t) - \boldsymbol{\psi}(\boldsymbol{z}_{1:t})\|_2^2\right]. \qquad (24)$$

Denote the minimizer of (24) at each $t$ by $\mathbf{W}_t^*$. Consider the situation that, at each $t$, we would like to predict $\boldsymbol{\psi}_t$ using a sequence of observations $\boldsymbol{x}_{1:t}$. Let $\boldsymbol{\sigma}(\cdot) \in \mathbb{R}^{K_\sigma}$ be sufficiently rich so that there exists a linear operator $\mathbf{W}_t^{(p)}$ that maps from the product space of $\boldsymbol{\sigma}(\boldsymbol{x}_1) \otimes \cdots \otimes \boldsymbol{\sigma}(\boldsymbol{x}_t)$ to $\boldsymbol{r}_t := \mathbb{E}_{q(\boldsymbol{z}_{1:t}|\boldsymbol{x}_{1:t})}[\boldsymbol{\psi}(\boldsymbol{z}_{1:t})]$, then $\mathbf{W}_t^{(p)}$ is an order $t+1$ tensor which is expensive to estimate. Low rank approaches may alleviate the difficulty, such as TT. In fact, the sequence of minimizers to (24)

$\{\mathbf{W}_{t'}^*\}_{t'=1}^t$ form a TT of an order $t+1$ tensor $\mathbf{W}_t^{(f)}$ with the same shape as $\mathbf{W}_t^{(p)}$. For example:

$$\mathbf{W}_1^* = \underset{\mathbf{W}_1}{\arg\min}\, \mathbb{E}_p \sum_i \left( \sum_j W_{1,ji}\sigma_i(\boldsymbol{x}_1) - \psi_{1,j} \right)^2 \Rightarrow r_{1,j} = \sum_{ji} \mathbf{W}_{1,ji}^* \sigma_i(\boldsymbol{x}_1)$$

$$\mathbf{W}_2^* = \underset{\mathbf{W}_2}{\arg\min}\, \mathbb{E}_p \sum_l \left( \sum_{jk} W_{2,lkj} r_{1,j}\sigma_k(\boldsymbol{x}_2) - \psi_{2,l} \right)^2$$

$$= \underset{\mathbf{W}_2}{\arg\min}\, \mathbb{E}_p \sum_l \left( \sum_{jk} W_{2,lkj} \left[ \sum_{ij} W_{1,ji}^* \sigma_i(\boldsymbol{x}_1) \right] \sigma_k(\boldsymbol{x}_2) - \psi_{2,l} \right)^2$$

$$= \underset{\mathbf{W}_2}{\arg\min}\, \mathbb{E}_p \sum_l \left( \sum_{ik} \underbrace{\left[ \sum_j W_{2,lkj} W_{1,ji}^* \right]}_{W_{2,lki}^{(f)}} \sigma_i(\boldsymbol{x}_1)\sigma_k(\boldsymbol{x}_2) - \psi_{2,l} \right)^2$$

the summation in the square brackets is the TT of $\mathbf{W}_2^{(f)}$. Thus, the proposed optimization for (24) finds a tensor of the same shape as $\mathbf{W}_t^{(p)}$ in the TT space sequentially, predicting a new $\psi_t$ by joining a new core tensor with $\mathbf{W}_{t\text{-}1}^{(f)}$, and only minimize the EMSE in the space of the new core tensor to get $\mathbf{W}_t^*$.

We argue that the computed $\boldsymbol{r}_t$ after sequentially optimizing $\mathbf{W}$ given a large set of training examples containing bootstrapped $\boldsymbol{r}_{t\text{-}1}$ does not diverge and produce a good approximation to the true posterior moments on $\psi_t$. For any $t$, the set of inputs in the regression contains $\boldsymbol{x}_t$, so if the regression is performed in closed-form rather than using the delta rule, the output $\boldsymbol{r}_t$ is at least close to the true $\mathbb{E}_{p(\boldsymbol{z}_{1:t}|\boldsymbol{x}_{1:t})}[\psi_t]$ as $\mathbb{E}_{q(\boldsymbol{z}_{1:t}|\boldsymbol{x}_t)}[\psi_t]$, the output of another regression similar to (5), which can be made closer to $\mathbb{E}_{p(\boldsymbol{z}_{1:t}|\boldsymbol{x}_t)}[\psi_t]$ using more flexible $\boldsymbol{h}$ and more training examples. At time $t+1$, the statistical dependency between $\boldsymbol{r}_t$ (depending on $\boldsymbol{x}_t$) and $\psi_{t+1}$ improves the prediction quality if $\boldsymbol{h}$ is flexible enough to pick up this dependency. At time $t+\tau$, as $\tau > 0$ increases, the prediction should continue to improve until $\boldsymbol{x}_t$ become uninformative of $\psi_{t+\tau}$, which depends on the range of temporal dependencies ("time constant") of the internal model and the encoding functions $\psi_t$.

# D   Experimental details

In all simulations in the main text, we assume the brain can draw samples from the internal model, and the recognition weights $\mathbf{W}$ and readout weights $\boldsymbol{\alpha}$ have been trained on these samples for a long time and have converged. In our experiments, this condition was achieved by closed-form regression in solving least square regressions, using 10,000-20,000 sequences from the internal model and a Tikhonov regularization on $\mathbf{W}$ with strength 0.001, and trained the recognition parameters for around 100 time steps in order for the SSM to enter in the stationary regime. The learned parameters are then fixed for online inference. The base tuning functions $\boldsymbol{\gamma}(\cdot)$ in (9) and input feature map $\boldsymbol{\sigma}(\cdot)$ in (12) and (13) have $\tanh$ nonlinearity after fixed random linear projections; the weights and biases in the projection are randomly drawn from a Gaussian with variance such that these functions are relatively smooth for the inputs they receive. Code is available at `https://github.com/kevin-w-li/ddc_ssm`

Figure 4: Example training data used for the auditory illusion experiment.

## D.1 Auditory continuity illusions

### D.1.1 Model setup

The internal model has a 2-D binary latent dynamics for the tone ($z_{t,0}$) and noise ($z_{t,1}$), and a 3-D noisy observation $x_{t,i}, i \in \{0, 1, 2\}$ for three frequency bands. Mathematically, it is defined as

$$c_{t,i} \sim \text{Bernoulli}(0.1) \qquad\qquad i \in \{0, 1\}$$
$$l_{t,0} \sim \text{Uniform}(\{2, 4\}) \qquad\qquad l_{t,1} \sim \text{Uniform}(\{1, 3\})$$
$$z_{t,i} = \begin{cases} z_{t\text{-}1,i} & \text{if } z_{t\text{-}1,i} \neq 0.0 \text{ and } c_{t,i} = 0 \\ c_{t,i}l_{t,i} & \text{if } z_{t\text{-}1,i} = 0.0 \\ 0 & \text{if } z_{t\text{-}1,i} \neq 0.0 \text{ and } c_{t,i} = 1 \end{cases} \qquad i \in 0, 2$$
$$x_{t,1} = \max\{z_{t,0}, z_{t,1}\} + \zeta_{t,i}, \qquad\qquad \zeta_{t,1} \sim \mathcal{N}(0, 0.1^2)$$
$$x_{t,i} = z_{t,1} + \zeta_{t,i}, \qquad\qquad \zeta_{t,i} \sim \mathcal{N}(0, 0.1^2) \qquad i \in 0, 2$$

In words, the tone has energy levels $\{0, 2, 4\}$ and the noise has energy levels $\{0, 1, 3\}$. At each time step, the tone and the noise can turn on or fall off with probability 0.1. For each of the two, if it turns on, it takes one of the two non-zero levels with equal chance; but it can only fall down to 0. The middle frequency channel reflects the greater level of the tone and the noise. The other two frequency channels only contain the noise. All three bands are contaminated by a small amount of i.i.d Gaussian noise. Example of the simulated data are shown in Figure 4.

In the DDC filter, we set the $K_\psi = 200$, $K_\gamma = 20$ and $K_\sigma = 10$ and used the $h^{bil}$ in (12).

### D.1.2 Additional methods and results

We showed in Figure 1 the marginal p.m.f of the inferred tone level given observations up to time the stimulus time $p(z_{t\text{-}\tau}|x_{1:t})$ decoded from $r_t$. This is done by first approximating posterior expectation over the static tuning function $\gamma(z_{t\text{-}\tau})$ (other choices of basis are possible) using (14), obtaining $m_{t\text{-}\tau} := \mathbb{E}_{qz_{t\text{-}\tau}|x_{1:t}}[Z_{t\text{-}\tau}|x_{1:t}]$, a DDC on $Z_{t\text{-}\tau}|x_{1:t}$. Using maximum entropy decoding, we can find the corresponding p.m.f. Let the discrete p.m.f be $p(z_{t\text{-}\tau}|x_{1:t}) = \prod_i^{|\mathcal{Z}|} p_i^{\delta(z_{t\text{-}\tau}=z_i)}$, where $|\mathcal{Z}|$ is the cardinality of the support on $z$ (9 in this case), and $\pi$ is the discrete probabilities that can be decoded from $r$ and $\gamma$ by solving the following optimization problem:

$$\min_{p} \sum_i^{|\mathcal{Z}|} p_i \log(p_i) \quad \text{s.t.} \sum_i^{|\mathcal{Z}|} p_i \gamma_j(z_i) = m_j, \sum_i^{|\mathcal{Z}|} p_i = 1, p_i \in [0, 1], \qquad (26)$$

which is relatively simple for a 9-outcome (3 tone $\times$ 3 noise levels), discrete distribution.

Figure 5: Maximum entropy decoding of the posterior marginals in the tracking experiment, compared with Figure 3 which is obtained by approximating expectation of bin functions.

## D.2 Flash-lag effect

The internal model that reproduced the smoothing effect is

$$p(\boldsymbol{z}_t|\boldsymbol{z}_{t\text{-}1}) = \mathcal{N}([A\boldsymbol{z}_{t\text{-}1}]_+, [0.01^2, 0.002^2, 1e^{-15}]) \qquad A = \begin{bmatrix} 0.0 & 1.0 & 0.0 \\ 0.0 & 0.0 & 1.0 \\ 0.0 & 0.0 & 0.8 \end{bmatrix} \qquad (27)$$

$$p(x_{t,i}|\boldsymbol{z}_t) = \text{Poisson}\left(3\exp\left[-\frac{(\text{loc}(i) - z_{t,0})^2}{2 \times 1.5^2}\right]\right) \qquad (28)$$

where $[]_+$ is a elastic bounding box at $\pm 1$. $loc$ is a linear transformation from pixel numbers to real values.

In the DDC filter, we set the dimensionalities $K_\psi = 500$, $K_\gamma = 100$ and $K_\sigma = 150$ and used the $\boldsymbol{h}^{lin}(\cdot)$ in (13).

## D.3 Noisy and occluded tracking

The internal model has 3-D latent (2 continuous, 1 discrete) and 30-D observation.

$$p(\boldsymbol{z}_t|\boldsymbol{z}_{t\text{-}1}) = \mathcal{N}(\boldsymbol{f}(\boldsymbol{z}_{t\text{-}1}), [0.1^2, 0.1^2]) \tag{29}$$

$$\boldsymbol{f}(\boldsymbol{z}_t) = s_t \boldsymbol{A} \boldsymbol{z}_{t\text{-}1} \tag{30}$$

$$s_t = \frac{1}{\|\boldsymbol{z}_{t\text{-}1}\|_2 \exp(-4(\|\boldsymbol{z}_{t\text{-}1}\|_2 - 0.3) + 1)} \tag{31}$$

$$p(m_t|m_{t-1}) = (\text{Bernoulli}(0.1) + m_{t-1}) \mod 2 \tag{32}$$

$$p(z_{t,i}|\boldsymbol{z}_t, m_t) = \mathcal{N}\left(\max\left\{\exp\left[-\frac{(\text{loc}(i) - z_{t,0})^2}{2 \times 3^2}\right], m_t\right\}, I_{30} 0.1^2\right) \tag{33}$$

$$\tag{34}$$

where $loc$ is a linear transformation from pixel number to real values, and $\boldsymbol{A}$ is a rotation matrix by $\pi/8$. Due to the sigmoidal scaling, $\boldsymbol{z}_t$ stays around the unit circle most of the time, but can occasionally cross through the origin due to noise.

In the DDC filter, we set the dimensionalities $K_{\boldsymbol{\psi}} = 500, K_{\boldsymbol{\gamma}} = 100$ and $K_{\boldsymbol{\sigma}} = 200$ and used the $\boldsymbol{h}_{\mathbf{W}}^{lin}(\cdot)$ in (13).

The histogram decoding from $\boldsymbol{r}_t$ is expected to be noisier due to the non-smoothness of the bin functions Figure 3, but still shows meaningful temporal integration of the observations. Results of the maximum entropy decoding of the posterior marginals are shown in Figure 5, which is less smooth due to $\boldsymbol{r}_t$ being not exactly in the set of feasible sufficient statistics.

## E  Robustness against neuronal noise

In the main text, we have discussed DDC when the representation $\boldsymbol{r}_{Z|\boldsymbol{x}}$ is deterministic, but real neurons are noisy. In this case, the spike count in some time window are taken as a noisy DDC representation. A noisy DDC may not identify a member in the class of exponential family distributions specified by $\boldsymbol{\gamma}$, as it may not correspond to any valid mean parameter. However, if the noise has zero mean (including Poisson noise) and no or weak correlation, then it does not fatally harm inference or learning to infer, as long as the training for $\mathbf{W}$ and $\boldsymbol{\alpha}$ is performed also on noisy DDC and on function evaluations on noisy samples. For inference, this type of noise on the input tends to average out in summations or inner products, the main operations in DDC computations as in (2) and (6). For learning to infer, noise on the input acts as a regularizer for $\mathbf{W}$, and noise on the output does not change the solution to regressions.

To verify our intuitions, we re-ran the experiments with the following changes to the DDC filter:

- The nonlinearity in $\boldsymbol{\gamma}(\boldsymbol{z}_t)$ and $\boldsymbol{\sigma}(\boldsymbol{x}_t)$ changes from $\tanh(\cdot)$ to $\text{sigmoid}(\cdot)$
- Independent Poisson noise is added to each feature evaluation of $\boldsymbol{\gamma}(\boldsymbol{z}_t)$, $\boldsymbol{\sigma}(\boldsymbol{x}_t)$ and $\boldsymbol{k}(\boldsymbol{r}_{t\text{-}1}, \boldsymbol{x}_t)$ and output of $\boldsymbol{h}_{\boldsymbol{\phi}}$.

and the results are shown in Figure 6 for the smoothing in flash-lag effect and Figure 7 for occluded tracking. The results are mostly the same as using noiseless DDC, although higher variability in prediction resulting from a noisy representation is clearly visible.

Figure 6: Same as Figure 2 but with noisy DDC. The error bars of DDC models are stds from 100 runs.

Figure 7: Same as Figure 3 but with noisy DDC.