[Reviews · NeurIPS 2019]

Reviewer 1



This paper addresses the problem of biologically-plausible perceptual inference in dynamical environments. In particular, it considers situations in which informative sensory information arrives delayed with respect to the underlying state and thus require ‘postdiction’ to update the inference of past states given new sensory observations. The authors extend a previously published method for inference in graphical models (DDC-HM) to temporally extended encoding functions and test their model in three situation cases where postdiction is relevant. Overall, I find this work valuable and interesting. It could, however, be more clearly presented and provide some relevant comparisons with alternative models. Regarding originality, the technical contributions, (1) an extension of the DDC-HM method to temporally extended (recurrent) encoding functions and (2) a way to train the model are original and nontrivial. In terms of clarity, the theory is well described but a more clear exposition of the algorithm would help (including an algorithm box). Moreover, the results could be more clearly presented - in particular the figures. Figure 1 is pretty difficult to parse. The relevant comparison (between panels B and C) takes significant time to decode, which could be solved by better/additional labels or annotations. The inset occupies much of the figure but is never referred to, I think. In Figure 2, I couldn’t find the gray line in panel A, maybe because of its poor resolution. Also, I failed to find the precise definition of t_0 and \tau in the internal model. Most importantly, in my opinion, the work lacks a comparison of the proposed model to other techniques in variational inference for dynamical environments (e.g..variational recurrent neural networks). Without taking biological plausibility into account, it would be very interesting to know if alternative models can account for the behavioral phenomena or whether this is a particular feature of the proposed method. Without that comparison it seems that the work is half done. Small details: L179 “simple internal model for tone and noise stimuli described by (19) in Appendix C.1.” In Fig 3, C-F panels are all labeled C. ===UPDATE=== I'd like to thank the authors for a detail and thoughtful response to the reviews. After reading the rebuttal, I'm overall pleased with the work. In my view, however, adding a comparison with another variational inference method (even one clearly not biologically plausible) would strengthen the paper's results. I have therefore chosen to maintain my score (6. Marginally above acceptance...).

Reviewer 2



After reading the Author Summary: The authors have clarified some of my doubts and addressed my concerns. Also, I find the discussion about signatures of internal representation quite interesting and I recommend to include it in the revised paper. Overall, I confirm my original judgment that this is an important contribution to NeurIPS and theoretical neuroscience. Summary: In this paper, the authors extend the theory of distributed distributional codes (DDC) to address the issue of how the brain might implement online filtering and postdiction. They provide three example applications (an auditory continuity illusion, the flash-lag effect, tracking under noise and occlusions) that qualitatively reproduce findings from behavioral studies. Originality: High. There is relatively little work on building dynamical Bayesian inference models that account for *postdiction* effects; and this is a novel extension and application of the recently proposed distributed distributional codes Quality: High. This is a thoughtful, well-researched paper which provides an interesting theoretical contribution and well-designed simulations. Clarity: The paper is well-written and well-structured and the images provide useful information. Significance: This is an interesting contribution both for the attention it brings to the phenomena of postdiction and for an application of the DDC. Major comments: This is a solid, interesting paper and a pleasure to read, and I congratulate the authors for making their code entirely available (although as a good practice I recommend to comment the code more, even in production). The main point of DDC is to provide a means to represent complex probability distributions, learn, and compute with them in a simple, biologically plausible way (namely, using only linear operations or the delta rule); and this paper shows how the DDC framework can be extended to deal with online filtering (and postdiction of the entire stimulus history). One question I have is with respect to the tuning functions \gamma. First, it might help to clarify in Section 2 that the tuning functions \gamma are given (and not learnt) in the DDC setup. Could they be learnt as well? Second, the current work uses tanh nonlinearities after random linear projections (as explained in the Supplementary Material; but I recommend to specify it as well in the main text, perhaps in Section 4). I gather that the results are relatively robust to changes in the set of \gamma (as long as it constitutes a "good" basis set), and did the authors check sensitivity of their results to different randomized basis sets? Another point about DDC is that they are not easily amenable to computations besides the calculation of expectations (the authors need to use a bunch of additional methods to compute the marginals and in particular the joint posteriors, as explained in the Supplementary Material) as opposed, e.g., to a popular "sampling" proposal. The fact that DDC provide "implicit" representations might be an advantage in explaining why humans are good at implicit computations of uncertainty but not necessarily at reporting them explicitly. Can the authors elaborate briefly on consequences/signatures of their model that would allow it to be distinguished from competing proposals for internal representations? Minor comments: line 153-154: "Given that there exists a minimum in (8), the minimum of (9) also exists for most well-behaved distributions on x1:t-1, and is attained if W minimizes (8) on all possible x1:t-1." Can the authors expand a bit on this? Clearly if W minimizes (8) on all possible paths (which seems a fairly strong assumption) then it also satisfies (9); but what matters is that a solution satisfying (9) does not necessarily imply (8), unless there is some additional property. (This is partially addressed in Section B of the Supplementary Material.) Figure 2A: I recommend to re-draw panel A based on the data in the original paper, because the current panel is unreadable. Typos: line 44: percpets --> percepts line 57: it is a challenge for the brain learns to perform inference --> unclear phrasing (if "for" is meant as "because", the subject is hanging - what is a challenge? otherwise, there must be a typo) line 61: analyitical --> analytical line 63: can be approximated the optimal solutions --> can approximate the optimal solutions line 66: that address --> that addresses line 83: easy to samples --> easy to sample (from) line 85: a full stop or semicolon is missing at the end of the line line 141: "time steps" missing at the end? (and full stop) line 150: recogntion --> recognition line 156: h_W^bil --> W should be italicized line 169: for each experiments --> for each experiment line 179: the internal model is described by Eq. 15 in the appendix? (not Eq. 19) Fig 1 caption: "showing decoded marginal distribution for perceived over tone level"? (probably perceived --> posterior) Fig 1 caption: marked form A to F --> marked from A to F Fig 2 caption: posdition --> postdiction line 245: bimodalality --> bimodality References: Please double-check (e.g., reference [37] is missing the authors). Fig 3 caption: estiamted --> estimated Supplementary Material: line 382: used to be approximated the conditional -> used to approximate the conditional line 384: one interpret --> one interprets line 386: Approximation solution --> Approximated solution line 418: the middle band of reflects... --> ? line 423: something is wrong at the end of the line line 433: one can decoding --> one can decode line 450: new observation --> the new observation

Reviewer 3



The authors propose a DDC as a framework for representing an inference network that describes a stochastic latent dynamical system. The authors claim that the DC framework is what makes this method “neurally plausible.” The authors describe their method in detail, describe a learning procedure, and demonstrate the utility of their method by showing that the method reproduces some psychophysical observations from human studies. The paper appears to be well researched and technically well developed but lacks somewhat in clarity. In particular, the development of the DDC framework and learning rules is very detailed, but the authors are a bit loose with language and its not clear from paragraph to paragraph where they are going. Since the DDC is not a standard approach to modeling and inference, it is the onus of the authors to hold the hand of the reader a bit. I outline many of these points below along with some grammatical and spelling errors. Equation (2): The authors do not distinguish what they mean by “neurons”. For example, in equation (2), are the authors meaning to describe the mean firing rate of biological neurons or activations of neurons in an artificial neural network? If it’s the latter, how does this relate to real neurons, where there is point-process noise? Line 92: the authors state “ encode a random variable” but equation (2) does not seem to be encoding a random variable but the distribution of that variable. Maybe this is a distinction without a difference but I find it helpful to be clear here. If I understand correctly r_Z is not a function of z, but is just a list of numbers that is a (possibly lossy) encoding of the distribution over Z. This list of numbers is then used to compute approximate expectations wrt that distribution. Is that correct? Line 118: “as long as the brain is able to draw samples according to the correct internal model” Isn’t the point to learn the correct model? This seems like circular reasoning. How is the brain to learn the model of a distribution by sampling from that distribution? The same goes for the sampling described in Section 3.2. Line 141: “for at least K_\psi/K_\gamma” at least K_\psi/K_\gamma what? This sentence seems to have been cut off abruptly. Eq (3) and (12): How do the authors propose that the brain learn the alphas? Should this be obvious? Should it be trivial that the brain knows what function it should approximate? Line 41-42: “however, the plausibility or otherwise of these hypotheses remains debated” Its not clear that the authors have improved on this state of affairs in this paper. The authors could comment further on this point. Lines 190-195: This section seems superfluous. Can the authors describe what relevance this section has to the literature? What is the point they are making here? Maybe this space could have been better used by fleshing out their method more carefully. Figure 1: Why are there multiple buses displayed at some time points? Line 46: “The causal process of realistic..”-> “The causal process of a realistic...” Line 49: “ analytical solution are…”->” analytical solution is...” Line 57: “for the brain learns…” ->“for the brain to learn...” Line 83: “ easy to samples” -> “easy to sample”

[Author Response · NeurIPS 2019]

Thanks for the comments. There were some suggested improvements – if accepted we will incorporate: an algorithm
box; analysis of complexity; redesign and/or annotation for Figure 1; Figure 2A adaptation; and typo corrections.

**R1: Fig 2**: We will delete the reference to a gray line, which had been removed from the figure – sorry. Parameters
$t_0$ and $\tau$ affect readout, not the internal model or inference: $t_0$ is the perceived time of the flash; $\tau$ is the postdictive
window – the duration of subsequent evidence that contributes to the estimate of the location of the moving object at $t_0$.
**Comparisons**: Biological plausibility is at the heart of the problem we seek to address. Postdictive behaviour depends
on updating beliefs about *past* states – the central question of our work is how this might happen in natural online
inference. Standard temporal variational methods (e.g. Chung et al., 2015; Fraccaro et al., 2017) represent states
individually, and so would require acausal ("backward") message passing to revise past states (see also comments
under Signatures below). This is possible (e.g. Fraccaro; or Johnson et al., 2016), but currently only implemented for
restricted classes of generative model which do not capture the natural environment.

**R2: Encoding functions**: indeed, we chose fixed but random $\gamma$. Results are robust to redrawing random projections,
and to different nonlinearities. To train the parameters of $\gamma$ and $\psi_t$ one could minimize the predictive error (5) or (9)
wrt these parameters. Gradient descent would be a natural choice, albeit with debatable biological plausibility.
**Signatures of internal representation**: a general attempt to distinguish encoding schemes that use expectations (DDC),
samples, or natural parameters (including the 'PPC') depends on additional model assumptions and is larger than can
be addressed satisfactorily in a NeurIPS paper. However, in the specific context of temporal models and postdiction,
these schemes differ in the mechanism by which beliefs about the past may be maintained and updated. The linearity of
expectation, combined with temporal encoding functions (or sufficient statistics) $\psi_t$, allows a simple dynamical update
rule to maintain and update beliefs (eqs (7) and (10) or (11)). If beliefs about latent states were represented by samples,
then these samples would need to be maintained in a sort of "buffer" to be modified by later inputs and accessed by
downstream processing. Natural parameter encodings could, in principle, exploit temporal sufficient statistic functions
similar to those we introduce, but the dynamical updates required to maintain and revise the natural parameters through
time would be far more complex to derive and implement than for the mean parameters of the DDC. Thus, of these
options, we believe the DDC inference and postdiction in the temporal context comes closest to the known biology.
**Lines 153-154**: (We will clarify the entire first paragraph of 3.2 if accepted.) Equation (8) would have been better
written: $\mathcal{L}^f(\widetilde{W}_t; x_{1:t\text{-}1}) = \mathbb{E}_{q(z_{1:t}, x_t | x_{1:t\text{-}1})} \left[ \| \widetilde{W}_t \sigma(x_t) - \psi(z_{1:t}) \|_2^2 \right]$. That is, the loss is history-dependent and the
optimal weights thus depend on both history and time. Now, if we restrict functions $h_W$ in (9) to be linear in $\sigma(x_t)$ (as
both (10) and (11) are) then $h_W$ can be seen as effectively mapping $r_{t\text{-}1}$ to a time- and history-dependent $\widetilde{W}_t$. Thus,
although the loss (9) is the expectation of (8) over histories, the optimal sequence of $\widetilde{W}_t$ for each history still provides a
target for the minimisation of (9), and the expectation of the minima of (8) bounds the minimum of (9) below. But
indeed, the constrained form of $h_W(r_{t\text{-}1}, \cdot)$ means that this lower bound will not generally be achieved.

**R3: "Neurons"**: Our intention is for activities of neurons in our scheme to provide a model for the firing rates of
biological neurons. We expect DDC-based computations to be robust to Poisson-like noise, as linear operations acting
on populations of neurons (inference and readout) effectively average away independent perturbations. Consistent with
this intuition, when we re-ran the flash-lag experiment using Poisson neurons the mean results were essentially the
same, albeit with greater trial-to-trial variance. For occluded tracking, the $R^2$ in postdicting the true $z_t$ by posterior
mean still increased with postdictive window $\tau$, although this improvement did not extend as far into the past as in the
noiseless case. If accepted, we will include these new experimental results.
**Line 92**: Indeed, DDC activities encode a *distribution* or *belief* about a random variable – we will amend the text.
**Line 118**: Our focus here is to propose how neural circuits may implement postdictive inference – itself, a non-trivial
problem even when the internal model of the world is known – and relate such inference to psychometric phenomena.
A general approach to modelling learning using DDC-based inference in wake-sleep has been proposed before (Vertes
and Sahani, 2018), although extensions to temporal models are the subject of current research.
**Learning the readout**: the key feature of (3) and (12) is that $\alpha$ can be found for known target functions using
evaluations of [e.g. for (12)] $\psi(z_{1:t})$ and $l(z_{t\text{-}\tau})$, and distributional uncertainty then propagates appropriately. Thus (as
with the recognition weights) we need only simulations of $z_{1:t}$ from the internal model, potentially from the sleep phase
of wake-sleep. More generally, the target function $l()$ might need to be learnt based on supervision or reinforcement.
This could follow "normal" learning rules; with the DDC again ensuring that uncertainty is handled correctly.
**Lines 190-195**: These simulations probe the quality of inference in the learnt model. Figure 1D reproduces the known
effect that perceptual continuity depends on the level of the interrupting noise. Figure 1E demonstrates quantitative
inference – the posterior on continuity is reduced when the noise is quieter still. Finally, 1F demonstrates that the
inference model has correctly encoded the fact that tones in the generative model do not change in loudness.
**Fig 1, multiple buses**: The hallmark of postdiction is that beliefs about past values are revised by new evidence. Thus,
we show beliefs at time step $i$ (column) conditioned on sensory experience up to various later times $j$ (row). That is, the
three boxes in the $i$th column of the $j$th row show decoded posterior probabilities for the three possible levels of the tone
at step $j$, based on sensory input up to time $i$. The smaller inset buses show the inferred noise level in the same way.

[Meta-Review · NeurIPS 2019]

After discussion, the reviewers and I agree that this paper is an interesting and novel extension of DDCs to nonlinear filtering and smoothing problems. The reviewers raised a number of areas for improvement -- it is important that these be addressed prior to publication. To highlight two, there were key places where clarity should be improved (esp. surrounding Figure 1 and the core inference/learning algorithms), and the framing relative to the long line of work on neurally plausible filtering and smoothing mechanisms should be made clear. Ideally, previous works would be included in the experimental comparisons.